# The Impact of Acute Kidney Injury in the Perioperative Period on the Incidence of Postoperative Delirium in Patients Undergoing Coronary Artery Bypass Grafting—Observational Cohort Study

**DOI:** 10.3390/ijerph17041440

**Published:** 2020-02-24

**Authors:** Katarzyna Kotfis, Justyna Ślozowska, Mariusz Listewnik, Aleksandra Szylińska, Iwona Rotter

**Affiliations:** 1Department of Anesthesiology, Intensive Therapy and Acute Intoxications, Pomeranian Medical University in Szczecin, Al. Powstańców Wielkopolskich 72, 70-111 Szczecin, Poland; justynagarlak@gmail.com; 2Department of Cardiac Surgery, Pomeranian Medical University in Szczecin, Al. Powstańców Wielkopolskich 72, 70-111 Szczecin, Poland; sindbaad@poczta.onet.pl; 3Department of Medical Rehabilitation and Clinical Physiotherapy, Pomeranian Medical University in Szczecin, ul. Żołnierska 48, 71-210 Szczecin, Poland; aleksandra.szylinska@gmail.com (A.S.); iwona.rotter@pum.edu.pl (I.R.)

**Keywords:** acute kidney injury, AKI, KDIGO, POD, delirium, creatinine, glomerular filtration rate, GFR, CABG

## Abstract

Recent data indicate that acute kidney damage leads to inflammation in the brain and other distant organs. The purpose of this study was to investigate the effect of acute kidney injury (AKI) according to the Kidney Disease Improving Global Outcome (KDIGO) criteria on the occurrence of postoperative delirium in patients undergoing coronary artery bypass grafting (CABG). We performed a retrospective cohort analysis that included all consecutive patients undergoing elective CABG. The CAM-ICU (Confusion Assessment Method for Intensive Care Unit) was used for delirium assessment. Patients were divided into four groups, depending on the occurrence of AKI in the perioperative period according to KDIGO criteria. Overall, 902 patients were included in the final analysis, the mean age was 65.95 ± 8.01 years, and 76.83% were males (693/957). The majority of patients presented with normal kidney function-baseline creatinine level of 0.91 ± 0.21 (mg/dL). The incidence of AKI in the perioperative setting was 22.17% (200/902). Postoperative delirium was diagnosed in 115/902 patients (12.75%). Compared with no AKI, the odds of developing POD were increased for KDIGO stage 1 (OR 2.401 (95% confidence interval 1.484–3.884), *p* < 0.001); KDIGO stage 2 (OR 3.387 (95% confidence interval 1.459–7.866), *p* = 0.005); and highest for KDIGO stage 3 (OR equal to 9.729 (95% confidence interval 2.675–35.382), *p* = 0.001). Acute kidney injury, based on AKI staging, should be regarded as an independent risk factor for postoperative delirium after cardiac surgery.

## 1. Introduction

Acute kidney injury (AKI) is common in the perioperative setting in patients undergoing cardiac surgery and in critically ill patients and can contribute to increased morbidity and mortality by affecting distant organ function [1,2,3]. Likewise, acute brain dysfunction is common after cardiac surgery and can be manifested as postoperative delirium (POD) or delirium in the intensive care unit (ICU Delirium) [4]. Similar to AKI, postoperative delirium can be associated with adverse short- and long-term effects, including postoperative cognitive dysfunction (POCD), and increased morbidity and mortality [5].

Postoperative delirium and other postoperative neurological disorders (e.g., coma) are common, yet they are underdiagnosed symptoms of brain dysfunction in the intensive care unit, and their mechanisms are insufficiently understood [6,7]. Experimental evidence indicates that acute kidney dysfunction or damage contributes to brain dysfunction, but there are few studies that were designed to investigate the relationship between acute kidney damage and brain dysfunction during a serious illness or major surgery. Available research indicates that acute kidney injury is associated with delirium during a critical illness after consideration for potential confounders [8,9,10]. This suggests that AKI and its consequences for homeostasis may be an underestimated element that is important in the pathogenesis of delirium.

Numerous studies have shown that acute kidney injury has a significant influence on the function of extrarenal organs, including the brain [11]. It has been reported that ischemic AKI may induce inflammatory response and even functional changes in the brain [3]. Initially, the so-called “kidney–brain crosstalk” was observed in chronic kidney disease [12]. There have been studies showing that, during AKI, the kidney and the brain interact through different mechanisms: cytokine-induced damage, sodium dysregulation, oxidative stress, extravasation of leukocytes, and by the use of water channels [12,13]. Moreover, it must be underlined that impaired renal function may lead to the reduced clearance of certain drugs, their metabolites, and/or neurotoxins. There are two studies in the literature addressing this problem in the ICU setting, but none in the cardiac surgery setting [8,9]. A study by Pisani et al. examined the relationship between kidney dysfunction and the occurrence of ICU delirium but did not differentiate between acute and chronic kidney disease [9]. Another study, performed by Siew et al., has reported that AKI is an independent risk factor for developing delirium or being diagnosed with coma during critical illness [8]. According to the authors of this paper, the use of renal replacement therapy has modified the peak serum creatinine and the development of delirium or coma [8]. 

The vast majority of cardiac surgery operations are performed with the use of extracorporeal circulation, which may cause major harm to renal performance. Scientific studies deliver conflicting results regarding the influence of cardiopulmonary bypass on development of acute kidney injury [13,14]. However, a possible correlation between the occurrence of acute kidney damage in the perioperative period or the need for continuous renal replacement therapy (hemofiltration), and the occurrence of postoperative delirium in patients undergoing cardiac surgery, can be attributed to the effect of cardiopulmonary bypass. To date, no work has been published on this subject in a group of cardiac surgery patients.

Therefore, the purpose of this study was to determine whether acute kidney damage in the perioperative period is associated with postoperative delirium in patients undergoing cardiac surgery and the impact of these disorders on early and long-term prognosis. We hypothesized that acute kidney damage during the perioperative course of cardiac surgery is independently associated with postoperative delirium.

## 2. Material and Methods

### 2.1. Study Group and Data Collection

We performed a retrospective cohort analysis that included all consecutive patients undergoing planned coronary artery bypass grafting (CABG) surgery at a university cardiac surgery department, between year 2014 and 2018. We excluded patients undergoing emergency surgery, with known cognitive impairment (diagnosis of dementia or cognitive dysfunction), patients without sequential serum creatinine measurements (preoperative and at least one postoperative result), with a diagnosis of chronic renal failure, patients who received chronic renal replacement therapy prior to surgery, and those with an initial estimated glomerular filtration rate (GFR) of less than 20 mL/min/1.73 m^2^. 

All demographic, medical, and laboratory data, including creatinine levels, were collected retrospectively from electronic medical records. All patients were assessed for the presence of postoperative delirium until the 5th day after surgery. The primary outcome was the presence of postoperative delirium. We analyzed the POD in relation to the difference between baseline serum creatinine before surgery and the maximal serum creatinine values within 48 h after surgery. 

### 2.2. Delirium Diagnosis

Mental state (normal vs. delirium) was assessed daily, using the Polish version of the ICU Confusion Assessment Method (CAM-ICU) and Richmond Agitation and Sedation scale (RASS) [5]. All patients were screened for delirium twice a day, by day and night shifts of doctors and nurses, and the diagnosis of delirium was made according to the DSM-5 criteria [15]. In addition to screening, the medical documentation was analyzed for descriptive diagnosis of delirium (medical and nursing descriptions, including an analysis of anti-delirious drug use).

### 2.3. Acute Kidney Injury Diagnosis

Patients were divided into 4 groups, depending on the occurrence of acute kidney injury in the perioperative period, diagnosed according to the KDIGO (Kidney Disease Improving Global Outcome) classification, using plasma creatinine concentration [16]. The following groups were established:

KDIGO 0 (no-AKI): maximum increase in serum creatinine after surgery <0.3 mg/dL or increase <50% relative to baseline creatinine before surgery.

KDIGO 1 (Stage 1): maximum increase in serum creatinine after surgery ≥0.3 mg/dL or increase ≥50% from baseline creatinine before surgery.

KDIGO 2 (Stage 2): maximum increase in serum creatinine after surgery equal to ≥100% increase in baseline serum creatinine.

KDIGO 3 (Stage 3): maximum increase in serum creatinine after surgery at the level of ≥200% compared to baseline serum creatinine or the need for renal replacement therapy.

### 2.4. Ethical Issues

The study was performed in accordance with the Declaration of Helsinki and Good Clinical Practice. It received a waiver from the Bioethical Committee of the Pomeranian Medical University due to its retrospective observational character (decision no. KB-0012/237/12/19).

### 2.5. Statistical Analysis 

The study group characteristics are presented, using the mean, standard deviation, and percentages. To evaluate the normality of the distribution of the studied variables, we used the Shapiro–Wilk test. The odds ratio (OR) of postoperative delirium was obtained through the use of univariable logistic regression. The results of regression were presented with the value of the odds ratio, with 95% confidence intervals and the statistical significance value. A *p*-value of <0.05 was regarded as statistically significant. To identify independent risk factors, we performed a multivariable analysis and included only those parameters that had a significance of *p* < 0.001 in the univariable analysis. Further analysis included KDIGO staging and the risk of POD development adjusted by age, sex, BMI, EF, and ESL. All data were analyzed, using licensed software Statistica 12 (StatSoft Inc., Tulsa, OK, USA). 

## 3. Results

Of the initial 1292 elective CABG patients, only 968 had a complete set of data regarding perioperative creatinine and were selected for analysis after applying inclusion and exclusion criteria. A further 55 patients were excluded due to the diagnosis of chronic renal failure, with 11 excluded due to chronic renal failure requiring chronic dialysis and/or with GFR <20 mL/min/1.73 m^2^. A total of 902 patients were included in the final analysis, as depicted by the study flowchart (Figure 1). 

Complete patients’ characteristics of the study population are shown in Table 1. The mean age of the population was 65.95 ± 8.01 years, and the majority of patients were males 693/957 (76.83%), with a mean BMI of 28.97 ± 4.24. The perioperative risk estimated by the EuroScore Logistic 2 (%) was 1.82 ± 1.57. Thirty-day mortality was 0.99% (9/902). Postoperative delirium was diagnosed in 115/902 patients (12.75%).

The majority of patients presented with normal kidney function prior to the operation, with the mean baseline creatinine level of 0.91 ± 0.21 and mean baseline GFR of 82.32 ± 16.02 (mL/min/1.73 m^2^). The incidence of AKI in the perioperative setting was 22.17% (200/902). Among AKI patients, 159/902 were classified as KDIGO 1 (17.63%) and 30/902 as KDIGO 2 (3.33%). Of the 200 patients with AKI, 11/902 received RRT and fulfilled the KDIGO 3 criteria (1.22%). The perioperative laboratory data and KDIGO classification are visible in Table 2.

The univariable analysis of risk factors for postoperative delirium development is shown in Table 3. Only values with *p* < 0.001 in univariable analysis were entered into the multivariable analysis.

Multivariable analysis has shown that, among the risk factors associated interdependently with the development of POD, only age (OR 1.078, *p* < 0.001) and AKI staged due to KDIGO classification are independent risk factor for developing POD in this group of patients (Table 4).

Compared with no AKI, the odds of developing postoperative delirium was statistically significantly increased in the setting of KDIGO stage 1 with OR 2.401 (95% confidence interval 1.484–3.884), *p* < 0.001; KDIGO stage 2 with OR 3.387 (95% confidence interval 1.459–7.866), *p* = 0.005; and highest for KDIGO stage 3 with OR equal to 9.729 (95% confidence interval 2.675–35.382), *p* = 0.001, as visible in Table 5 and Figure 2.

## 4. Discussion

We performed an analysis of a large cohort of patients undergoing planned coronary artery bypass grafting procedure and found that acute kidney injury, regardless of its staging according to KDIGO criteria, is associated with an increased incidence of postoperative delirium. After adjusting for potential confounders, KDIGO stage 1 doubles the risk, stage 2 triples the risk, and stage 3 increases the risk of delirium nine times. Our results indicate that acute kidney injury and acute brain dysfunction may be regarded as practical confirmation of the kidney–brain crosstalk. No previous study has been presented to prove this relationship in patients undergoing cardiac surgery. 

The strength of our study is in the homogeneity of the population under analysis. We included patients undergoing only one type of cardiac surgery, with a relatively low potential for triggering acute kidney injury, without previous diagnosis of chronic renal failure, with normal mean preoperative creatinine level and glomerular filtration rate. We used the KDIGO classification to stage acute kidney injury, because the definition of stage 1 AKI according KDIGO is very sensitive, with small, short increases in creatinine classified as acute kidney damage [16,17,18].

The results of our study are in line with the reports provided by other authors but performed in the ICU setting. Although they used different AKI criteria, Pisani et al. were the first to demonstrate that acute changes in kidney function, diagnosed as an admission creatinine level above 2 mg/dL, were associated with the development of ICU delirium [9]. Siew et al. have found that moderate and severe AKI was strongly associated with the diagnosis of delirium and coma in critically ill patients, with a protective effect of renal replacement therapy on brain function [8]. The authors provided the proof that, regardless of acute kidney injury diagnosis, whether based on KDIGO staging or on peak serum creatinine, the AKI is associated with delirium. Moreover, Siew et al. regarded AKI as a modifiable risk factor for delirium in the ICU. The definite difference between our analysis and the one provided by Siew et al. is the baseline creatinine level. Siew et al. used an estimated baseline creatinine level, whereas our study included only those patients with a reliable serum creatinine level result, performed directly (within 24 h), prior to the operation.

The mechanism of the interplay between acute kidney injury and acute brain dysfunction called the “brain–kidney crosstalk” has not been fully elucidated [19,20]. The underlying pathophysiology of the comorbid neurological disorders in kidney disease may be governed by common anatomic and vasoregulatory systems, as well as by humoral and non-humoral bidirectional pathways that influence both the kidney and the brain. During AKI, both organs might interact through multiple mechanisms: amplification of cytokine-induced damage, leukocyte extravasation, elements of oxidative stress, and dysregulation of sodium, potassium, and water channels [3]. Animal data suggest that acute kidney injury in an experimental setting has been associated with neuroinflammation in those brain domains that are associated with the development of delirium, namely the hippocampus and the cerebral cortex [3,12,21].

It has been shown that, after experimental renal ischemia, distant organ changes occur that include cytokine induction, leukocyte infiltration, and apoptotic cell death. The effect of AKI on distant organs has been shown by an elegant study by Grigorieyev et al., who tested the hypothesis that AKI may lead to a vigorous inflammatory response and produces distinct genomic signatures in the brain and lungs [11]. The authors have shown, in a murine model of 60 min bilateral kidney ischemia, that global transcriptomic changes and histologic injury occurs. These changes were evident at both early (6 h) and late (36 h) timepoints after AKI. The inflammatory transcriptome (including 109 genes) of both organs changed with marked similarity. This included the innate immunity genes, namely Cd14, Socs3, Saa3, Lcn2, and Il1r2. A further functional genomic analysis of these genes suggested that both IL-10 and IL-6 signaling was involved in the distant process of local inflammation, and this was supported by increased serum levels of IL-10 and IL-6 after ischemia-reperfusion [11]. 

Liu et al. have reported short-term effect of ischemic AKI on inflammatory and functional changes of the brain in mice [3]. The authors induced bilateral renal ischemia for 60 min and studied mice brains after 24 h. Compared with sham mice, those mice which developed AKI presented with neuronal pyknosis and microgliosis in the brain. Acute kidney injury also led to increased levels of the proinflammatory chemokines, keratinocyte-derived chemoattractant, and the G-CSF in the cerebral cortex and hippocampus. Moreover, an increased expression of glial fibrillary acidic protein was seen in the astrocytes of corpus callosum and the cortex. In addition, an extravasation of Evans blue dye into the brain tissue suggested that the blood–brain barrier was disrupted in mice with AKI [3]. Distant effects of acute kidney injury may have systemic effects and add to morbidity and mortality observed clinically, including kidney–brain crosstalk and cardiorenal syndrome [21,22].

As the study presented by us is of observational character, the data showed a strong correlation only between the level of AKI and the incidence of postoperative delirium. However, it is not evidence of a direct interplay between the kidney and the brain. There may be, but it will be extrapolating by saying the data suggest that the kidney is affecting the brain or vice versa. Further studies are necessary to investigate this observation.

Our study is not without limitations. First, this is a single-center observation; therefore, its generalizability may be limited. Second, in this analysis, we used only the serum creatinine change over time as the diagnostic criterion for AKI. Based on the KDIGO criteria, the acute kidney damage may be diagnosed according to the “creatinine” criterion (i.e., increase by 0.3 mg/dL (26.4 mmol/L) or percentage increase in serum creatinine greater than or equal to 50% (1.5-fold from baseline)) or the “diuresis” criterion (i.e., reduced urine output to less than 0.5 mL/kg per hour for more than six hours in 48 h). Many researchers believe that the creatinine criterion should be combined with the diuresis criterion for greater diagnostic sensitivity; therefore, the lack of the “diuresis” criterion in our study may be regarded as a limitation. Third, only creatinine was available as a biomarker of kidney dysfunction. In the future studies, it is worth considering using other markers for early identification of acute kidney damage, such as NGAL or cystatin C [23].

## 5. Conclusions

This large observational study provides data that acute kidney injury, based on AKI staging, should be regarded as an important and independent risk factor for postoperative delirium after cardiac surgery. Acute kidney injury stage 1, according to KDIGO, doubles the risk, KDIGO stage 2 triples the risk, and KDIGO stage 3 increases the risk of delirium by a factor of nine, after adjusting for potential confounders. All patients should undergo preoperative and postoperative creatine-level assessment to stage AKI, as AKI staging has been shown to be independently associated with delirium. Further studies are necessary to elucidate the underlying mechanism of acute brain dysfunction during an acute kidney disorder.

## Figures and Tables

**Figure 1 ijerph-17-01440-f001:**
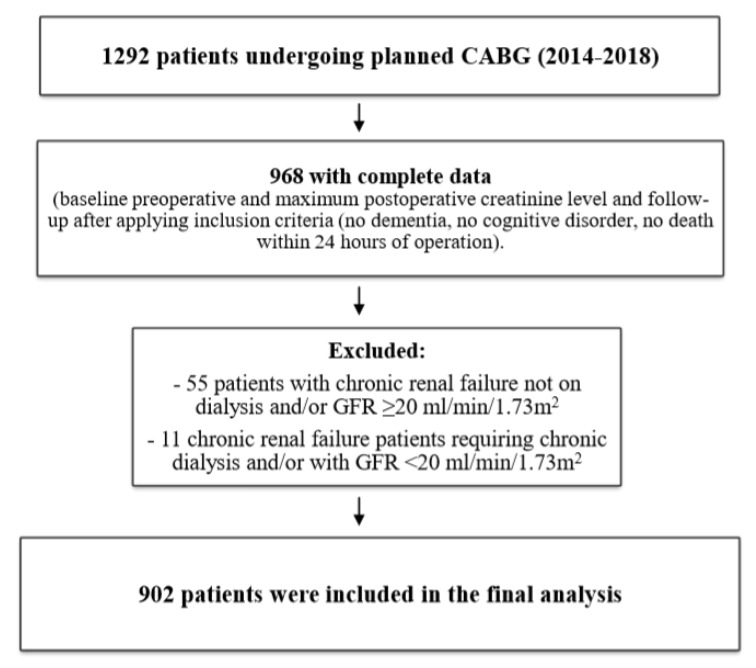
Study flowchart.

**Figure 2 ijerph-17-01440-f002:**
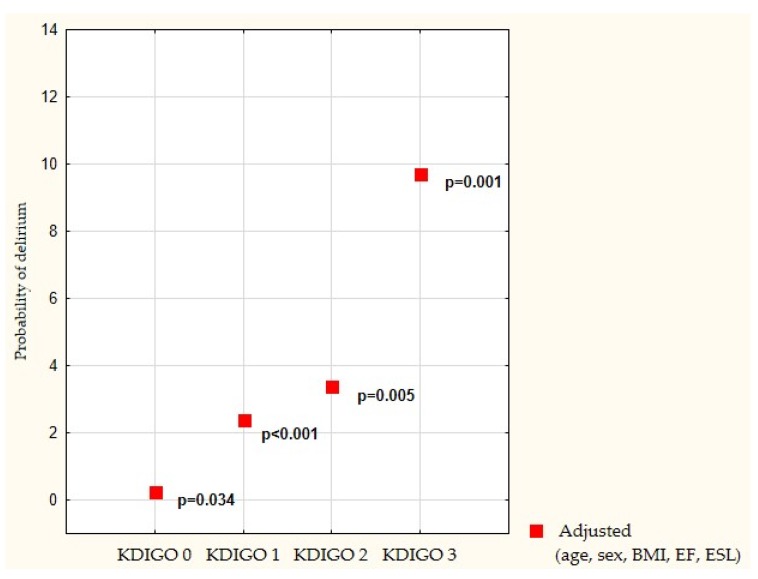
Risk of delirium according to adjusted and adjusted KDIGO staging. Legend: KDIGO—Kidney Disease Improving Global Outcome, BMI—body mass index, EF—Ejection fraction, and ESL—EuroScore Logistic 2.

**Table 1 ijerph-17-01440-t001:** Study-group characteristics.

Variable	Study Group (*n* = 902)
Age (years), mean ± SD	65.95 ± 8.01
Sex, male, *n* (%)	693 (76.83)
BMI (kg/m^2^), mean ± SD	28.97 ± 4.24
Ejection fraction, mean ± SD	47.57 ± 9.71
EuroScore Logistic 2 (%), mean ± SD	1.82 ± 1.57
**Concomitant diseases**	
Current smoking, *n* (%)	133 (14.75)
Arterial hypertension, *n* (%)	680 (75.39)
Myocardial infarction, *n* (%)	467 (51.77)
Heart failure (NYHA III or IV), *n* (%)	73 (8.09)
Atrial Fibrillation, *n* (%)	91 (10.09)
Diabetes [insulin], *n* (%)	97 (10.75)
Diabetes [oral medications/diet], *n* (%)	324 (35.92)
Impaired glucose tolerance, *n* (%)	31 (3.44)
Previous stroke, *n* (%)	40 (4.43)
Transient Ischemic Attack, *n* (%)	8 (0.89)
Internal Carotid Artery stenosis, *n* (%)	66 (7.32)
Extracardiac atherosclerosis, *n* (%)	156 (17.29)
Chronic Obstructive Pulmonary Disease, *n* (%)	41 (4.54)
**Perioperative data**	
Number of anastomoses, mean ± SD	3.07 ± 0.76
Crystalloid cardioplegia, *n* (%)	886 (98.23)
Perfusion time (min), mean ± SD	49.52 ± 13.51
Cross-clamping time (min), mean ± SD	29.89 ± 8.94
Intraoperative hemofiltration, *n* (%)	141 (15.63)
Intra-Aortic Balloon Pump, *n* (%)	1 (0.11)
Postoperative drainage, mean ± SD	486.99 ± 395.48
Reoperation (early), *n* (%)	32 (3.55)
**Postoperative outcome**	
Intubation time (min), mean ± SD	742.66 ± 858.44
Postoperative delirium, *n* (%)	115 (12.75)
Hospital LOS (days), mean ± SD	8.34 ± 6.29
ICU LOS (days), mean ± SD	2.43 ± 2.92
Mortality until 30 days, *n* (%)	9 (0.99)
Mortality above 30 days, *n* (%)	33 (3.66)

Legend: BMI—body mass index, LOS—length of stay, NYHA—New York Heart Association, *n*—number of patients, and SD—standard deviation.

**Table 2 ijerph-17-01440-t002:** Perioperative laboratory data and Kidney Disease Improving Global Outcome (KDIGO) classification.

Perioperative Laboratory Data	Study Group (*n* = 902)
Baseline CKMB, mean ± SD	25.86 ± 28.63
Maximum postop CKMB, mean ± SD	58.24 ± 64.62
Baseline serum creatinine, mean ± SD	0.91 ± 0.21
Maximum postop serum creatinine, mean ± SD	1.13 ± 0.49
Baseline GFR (mL/min/1.73 m^2^), mean ± SD	82.32 ± 16.02
Minimal postop GFR (mL/min/1.73 m^2^), mean ± SD	70.81 ± 22.39
No AKI according to KDIGO, *n* (%)	702 (77.83)
Any AKI according to KDIGO, *n* (%)	200 (22.17)
KDIGO Stage 1, *n* (%)	159 (17.63)
KDIGO Stage 2, *n* (%)	30 (3.33)
KDIGO Stage 3, *n* (%)	11 (1.22)
Postoperative CRRT, *n* (%)	11 (1.22)

Legend: AKI—acute kidney injury, CKMB—phosphocreatine kinase, CRRT—continuous renal replacement therapy, GFR—glomerular filtration rate, KDIGO—Kidney Disease Improving Global Outcome, *n*—number of patients, and SD—standard deviation.

**Table 3 ijerph-17-01440-t003:** Univariable analysis for POD development after CABG.

	Delirium			
	*p*-Value	OR	CI OR −95%	CI OR +95%
Age	<0.001	1.108	1.078	1.139
BMI	0.383	0.979	0.935	1.026
Ejection Fraction	0.372	0.991	0.971	1.011
EuroScore Logistic 2 (%)	<0.001	1.303	1.180	1.440
Number of anastomoses	0.435	1.108	0.857	1.432
Perfusion time	0.082	1.012	0.998	1.026
Cross-clamping time	0.041	1.022	1.001	1.044
Intubation time	0.638	1.000	1.000	1.000
Baseline CKMB	0.018	1.006	1.001	1.011
Maximum postop CKMB	0.079	1.002	1.000	1.004
Postoperative Drainage	0.240	1.000	1.000	1.001
Hospital LOS	<0.001	1.069	1.036	1.104
ICU LOS	0.215	1.031	0.983	1.081
Glycated hemoglobin HbA1c	0.001	1.306	1.108	1.538
Baseline creatinine	0.003	3.362	1.502	7.527
Baseline GFR (mL/min/1.73 m^2^)	<0.001	0.969	0.958	0.980
Maximum postop creatinine	<0.001	2.907	2.087	4.049
Minimal postop GFR (mL/min/1.73 m^2^)	<0.001	0.967	0.959	0.976
Sex, female	0.015	1.696	1.108	2.598
Current smoking	0.166	0.641	0.342	1.202
Heart failure NYHA III and IV	0.006	2.264	1.265	4.053
Previous stroke	0.360	1.481	0.639	3.431
Transient Ischemic Attack	0.310	2.304	0.459	11.554
Impaired glucose tolerance	0.296	0.463	0.109	1.965
Diabetes (oral medications/diet)	0.072	1.440	0.969	2.142
Diabetes (insulin)	0.244	1.409	0.792	2.506
Arterial hypertension	0.018	1.886	1.112	3.196
Myocardial infarction	0.060	1.466	0.984	2.184
Atrial fibrillation	0.428	1.278	0.697	2.344
Internal Carotid Artery stenosis	<0.001	2.857	1.597	5.111
Chronic Obstructive Pulmonary Disease	0.026	2.323	1.107	4.875
Extracardiac atherosclerosis	<0.001	2.173	1.389	3.399
Intraoperative hemofiltration	<0.001	2.530	1.610	3.977
Reoperation (early)	0.522	0.700	0.210	2.336
KDIGO Stage 1	<0.001	3.300	2.106	5.172
KDIGO Stage 2	<0.001	5.254	2.353	11.731
KDIGO Stage 3	<0.001	12.610	3.739	42.522

Legend: CKMB—phosphocreatine kinase, GFR—glomerular filtration rate, ICU—Intensive Care Unit, KDIGO—Kidney Disease Improving Global Outcome, LOS—length of stay, NYHA—New York Heart Association, and OR—odds ratio.

**Table 4 ijerph-17-01440-t004:** Multivariable analysis for POD development after CABG.

	*p*-Value	OR	CI OR −95%	CI OR +95%
Age	<0.001	1.078	1.046	1.111
Internal Carotid Artery stenosis	0.123	1.835	0.848	3.972
Extracardiac atherosclerosis	0.303	1.369	0.753	2.486
EuroScore Logistic 2 (%)	0.060	1.118	0.995	1.255
Intraoperative hemofiltration	0.047	1.683	1.008	2.810
KDIGO Stage 1	0.002	2.165	1.329	3.526
KDIGO Stage 2	0.033	2.619	1.079	6.356
KDIGO Stage 3	0.003	7.241	1.960	26.758
Hospital LOS	0.021	1.037	1.006	1.070

Legend: OR—odds ratio, KDIGO—Kidney Disease Improving Global Outcome, and LOS—length of stay.

**Table 5 ijerph-17-01440-t005:** Unadjusted and adjusted KDIGO staging and the risk of POD development.

		Delirium Unadjusted				Delirium Adjusted (Age, Sex, BMI, EF, and ESL)			
		*p*-Value	OR	CI OR −95%	CI OR 95%	*p*-Value	OR	CI OR −95%	CI OR 95%
KDIGO	Stage 1	<0.001	3.300	2.106	5.172	<0.001	2.401	1.484	3.884
	Stage 2	<0.001	5.254	2.353	11.731	0.005	3.387	1.459	7.866
	Stage 3	<0.001	12.610	3.739	42.522	0.001	9.729	2.675	35.382

Legend: OR—odds ratio; KDIGO—Kidney Disease Improving Global Outcome.

## Data Availability

The dataset used during the current study is available from the corresponding author upon reasonable request.

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
