# Peer review of "The Impact of Acute Kidney Injury in the Perioperative Period on the Incidence of Postoperative Delirium in Patients Undergoing Coronary Artery Bypass Grafting—Observational Cohort Study"

_ijerph, 2020, doi:10.3390/ijerph17041440_

Round 1

Reviewer 1 Report

General:

        The overall goal is to investigate whether acute kidney injury increases the incidence of postoperative delirium (POD) in patients undergoing coronary artery bypass grafting (CABG). It is a retrospective cohort analysis of 902 CABG patients, divided into 4 groups depending on the stages of AKI according to KDIGO criteria in the perioperative period. The results showed that KDIGO stage 1 doubled the risk, KDIGO stage 2 tripled the risk and KDIGO stage 3 increased the risk of POD by a factor of 9. The results indicated that AKI staging is associated with POD.

Specific comments:

This study has not investigated possible acute brain dysfunction resulting in POD after an acute kidney injury.

Author Response

Response: Thank you for this comment. We would like to point out that our observation is merely an association. We added the following sentence at the end of the discussion section: “As the study presented by us is of observational character, the data only showed a strong correlation between the level of AKI and the incidence of postoperative delirium. However, it is not an evidence of a direct interplay between the kidney and the brain. There may be, but it will be extrapolating by saying the data suggest kidney is affecting the brain or vice versa. Further studies are necessary to investigate this observation.”

Reviewer 2 Report

The authors, Kotfis et al, performed a large observational cohort study in order to identify correlation between Acute Kidney Injury and incidence of delirium. They analyzed >900 patients and demonstrated that OR of delirium positively correlated with the KDIGO score, through measuring the level of creatinine.

The study is good and very interesting to read. There are strong evidence presented to believe AKI is associated with delirium, which warrants further work. I recommend the manuscript be accepted with a few minor suggestions.

  1. Method: 2.3 AKI diagnosis - "plasma creatinine" was used in KDIGO 0 and 1, while "serum creatinine" was used in KDIGO2,3. Please keep it consistent.
  2. I would encourage the authors to add into the discussion that, as it is an observational study, the data only showed strong correlation between level of AKI and incidence of delirium. However, it is not an evidence of direct interplay between kidney and brain. There may be, but it will be extrapolating by saying the data suggest kidney is affecting the brain or vice versa.

Author Response

I recommend the manuscript be accepted with a few minor suggestions.

  1. Method: 2.3 AKI diagnosis - "plasma creatinine" was used in KDIGO 0 and 1, while "serum creatinine" was used in KDIGO2,3. Please keep it consistent.

Response: Thank you for this comment. This has been corrected to ‘serum’ for KDIGO 0 and 1.

  1. I would encourage the authors to add into the discussion that, as it is an observational study, the data only showed strong correlation between level of AKI and incidence of delirium. However, it is not an evidence of direct interplay between kidney and brain. There may be, but it will be extrapolating by saying the data suggest kidney is affecting the brain or vice versa.

Response: Thank you for this suggestion. We added the following sentence at the end of the discussion section: “As the study presented by us is of observational character, the data only showed a strong correlation between the level of AKI and the incidence of postoperative delirium. However, it is not an evidence of a direct interplay between the kidney and the brain. There may be, but it will be extrapolating by saying the data suggest kidney is affecting the brain or vice versa. Further studies are necessary to investigate this observation.”

Reviewer 3 Report

This paper describes the association of acute kidney injury and post-operative delirium in patients undergoing coronary artery nypass grafts. It is well done and analysed. There are no major criticisms.

Figure 5 repeats the data from Table 5 and is unnecessary. I am also not sure that the lines drawn between the various stages of AKI (ie. KDIGO 0, KDIGO 1, etc) are valid as these are discontinuous variables.

In the Discussion on page 9, the statement that acute kidney injury and acute brain dysfunction ”share a common pathway” is meaningless and should be removed.

The English is not bad but does require some refinement. For example, in the Abstract;

Line 6: AKI in the perioperative

Line 8: The majority

Line 9: The incidence

Line 11: no AKI, the odds

Author Response

This paper describes the association of acute kidney injury and post-operative delirium in patients undergoing coronary artery bypass grafts. It is well done and analysed. There are no major criticisms.

 Thank you.

Figure 5 repeats the data from Table 5 and is unnecessary. I am also not sure that the lines drawn between the various stages of AKI (ie. KDIGO 0, KDIGO 1, etc) are valid as these are discontinuous variables.

 Response: Thank you for this comment. We understand this point of view, however we would like to keep Figure 2 as it does not contain all the information from Table 5. In our opinion it is a good visualization of the results. We removed the lines from Figure 2 and inserted a modified version of the Figure into the manuscript.

In the Discussion on page 9, the statement that acute kidney injury and acute brain dysfunction ”share a common pathway” is meaningless and should be removed.

Response: Thank you. This has been removed.

The English is not bad but does require some refinement. For example, in the Abstract;

Line 6: AKI in the perioperative

Line 8: The majority

Line 9: The incidence

Line 11: no AKI, the odds

Response: Thank you for this comment, we agree and this has been corrected. The English in the abstract was not perfect as there is a word limit for the abstract. The current version of the abstract now reads:

Abstract: Recent data indicates that acute kidney damage leads to inflammation in the brain and other distant organs. The purpose of this study was to investigate the effect of acute kidney injury (AKI) according to KDIGO criteria on the occurrence of postoperative delirium in patients undergoing coronary artery bypass grafting (CABG).We performed a retrospective cohort analysis including all consecutive patients undergoing elective CABG. CAM-ICU was used for delirium assessment. Patients were divided into 4 groups depending on the occurrence of AKI in the perioperative period according to KDIGO criteria.Overall, 902 patients were included in the final analysis, the mean age was 65.95±8.01 years, 76.83% were males (693/957). The majority of patients presented with normal kidney function - baseline creatinine level of 0.91±0.21 [mg/dl]. The incidence of AKI in the perioperative setting was 22.17% (200/902). Postoperative delirium was diagnosed in 115/902 patients (12.75%). Compared with no AKI, the odds of developing POD were increased for KDIGO stage 1 (OR 2.401 [95% confidence interval 1.484-3.884], p<0.001); KDIGO stage 2 (OR 3.387 [95% confidence interval 1.459-7.866], p=0.005) and highest for KDIGO stage 3 (OR equal to 9.729 [95% confidence interval 2.675-35.382], p=0.001). Acute kidney injury, based on AKI staging, should be regarded as an independent risk factor for postoperative delirium after cardiac surgery.

We have also corrected English in other parts of the manuscript. This is visible with a ‘track changes’ option.